



# Inclusion of a cold hardening scheme to represent frost tolerance is essential to model realistic plant hydraulics in the Arctic-Boreal Zone in CLM5.0-FATES-Hydro

Marius S. A. Lambert[1], Hui Tang[2,3], Kjetil S. Aas[2], Frode Stordal[2], Rosie A. Fisher[4], Yilin Fang[5], Junyan Ding[5] and Frans-Jan W. Parmentier[1,6]

[1]Centre for Biogeochemistry in the Anthropocene, Department of Geosciences, University of Oslo, 0315 Oslo, Norway
[2]Department of Geosciences, University of Oslo, 0315 Oslo, Norway
[3]Geo-Ecology Research Group, Natural History Museum, University of Oslo, 0562 Oslo, Norway
[4]CICERO - Center for International Climate Research, 0318 Oslo, Norway
[5]Pacific Northwest National Laboratory, Richland, WA, USA
[6]Department of Physical Geography and Ecosystem Science, Lund University, 223 62 Lund, Sweden

*Correspondence to*: Marius S. A. Lambert (marius.lambert@geo.uio.no)

**Abstract.** As temperatures decrease in autumn, vegetation of temperate and boreal ecosystems increases its tolerance to freezing. This process, known as hardening, results in a set of physiological changes at the molecular level that initiate modifications of cell membrane composition and the synthesis of anti-freeze proteins. Together with the freezing of extracellular water, anti-freeze proteins reduce plant water potentials and xylem conductivity. To represent the responses of
vegetation to climate change, land surface schemes increasingly employ 'hydrodynamic' models that represent the explicit fluxes of water from soil and through plants. The functioning of such schemes under frozen soil conditions, however, is poorly understood. Nonetheless, hydraulic processes are of major importance in the dynamics of these systems, which can suffer from e.g. winter 'frost drought' events.

In this study, we implement a scheme that represents hardening into CLM5.0-FATES-Hydro. FATES-Hydro is a plant
hydrodynamics module in FATES, a cohort model of vegetation physiology, growth and dynamics hosted in CLM5.0. We find that, in frozen systems, it is necessary to introduce reductions in plant water loss associated with hardening to prevent winter desiccation. This work makes it possible to use CLM5.0-FATES-Hydro to model realistic impacts from frost droughts on vegetation growth and photosynthesis, leading to more reliable projections of how northern ecosystems respond to climate change.





## 1 Introduction

In the arctic-boreal region, winters are dark, typically last over 6 months, and average temperatures are low (generally around -20°C). To survive the harsh winters of this region, cold-adapted vegetation goes through a set of physiological and structural changes to avoid desiccation and frost mortality (Bansal et al., 2016; Chang et al., 2021). This capacity of plants to withstand freezing can be highly variable throughout the year – depending on the species and environmental factors such as light and temperature. In summer, the tolerance to freezing is low, but it slowly increases when exposed to a decrease in photoperiod and temperature (Beck et al., 2004) — a process known as cold acclimation (Li et al., 2004; Wisniewski et al., 2018). Cold-acclimation triggers multiple physiological and biochemical responses which enable plants to i) tolerate extracellular ice formation and the resulting cellular dehydration (Levitt, 1980, Janská et al., 2010),  ii) avoid the formation of interstitial ice crystals by keeping tissues isolated from the cold and/or by regulating nucleation. Ice nucleation can be suppressed down to -38°C if biological antifreeze proteins such as dehydrins and flavonoids are synthetized (Hanin et al., 2011) – a process called supercooling. Between -20°C and -30°C, the formation of intracellular glass (vitrification) further enables cold acclimated woody plants to develop a resistance to much lower temperatures. Trees in northern latitudes have evolved resistance to naturally occurring temperatures as low as -70°C (Sakai, 1983), whereas herbaceous plants can rarely tolerate < -25°C. Experiments have demonstrated that trees can resist the temperature of liquid nitrogen (-196°C) when regulating ice nucleation (Rinne et al., 1998).

Plants exposed to temperatures below their tolerance threshold (which varies as a function of time and environmental conditions) suffer from 1) mechanical stress, due to the intercellular ice crystallization and fragility of the tissues, and 2) osmotic dehydration due to the freezing of intercellular water. Therefore, the increased survival of plant cells and tissues is the evolutionary benefit of cold acclimation. The two strategies of tolerance and avoidance are often found simultaneously in the same plant. Cold acclimation strategies, rates and ranges involve complex species-dependent physiological and gene expression changes (Chinnusamy et al., 2006; Welling & Palva, 2006).

The physiological adaptations induced by cold acclimation have a large impact on the hydraulics of plants. Apoplastic ice formation and the increased viscosity of protein-bound water particles are known to reduce water flow (Dowgert & Steponkus, 1984; Gusta et al., 2005; Smit-Spinks et al., 1984). In addition, water stress resulting from deeply frozen soils with low soil water potentials triggers slow developmental changes in the root structure to increase their impermeability and reduce water loss (Beck et al., 2007; Kreszies et al., 2019; North & Nobel, 1995). Other strategies include the creation of an air gap during root shrinkage (Carminati et al., 2009; North & Nobel, 1997) and the inhibition of conducting channels (Knipfer et al., 2011; Lee et al., 2005; Ye & Steudle, 2006). Since water transport is prohibited in cold-acclimatized tissues, growth is also affected and it is important that plants de-acclimate rapidly in spring to re-activate their metabolism.

Apart from the critical role of plant hydraulics for the survival of plants during droughts, it is also a major driver of species distribution (Fontes, 2019; Lopez-Iglesias et al., 2014; Navarro et al., 2019; Rasztovits et al., 2014). Species distribution is shaped by mortality and recruitment, and the ability of species to survive under periods of low water availability depends on





their hydraulic traits (Greenwood et al., 2017). Tolerance to and delay of desiccation, for example, involve traits such as greater

resistance to embolism (i.e. interruption of water flow through plant conductive vessels), the ability of cells to remain alive at low water levels and the reduction of water loss (Kursar et al., 2009; Tyree et al., 2003).

Despite its importance, terrestrial biosphere models typically rely on simple approaches to represent plant hydraulics, involving an extension of Darcy's law. Darcy stated that the flux of water anywhere in the soil plant continuum was proportional to the hydraulic conductivity and the water potential gradient (Darcy, 1856). One of the simplest implementations of this is to model

soil-root conductance as a function of soil moisture (Jarvis and Morison, 1981; Bonan et al., 2014). In more complex models, variable xylem conductance and water potential are added (Williams et al., 2001; Duursma and Medlyn, 2012). One of the most advanced approaches is the continuous porous media approach (Edwards et al., 1986; Sperry et al., 1998; Christoffersen et al., 2016; Mirfenderesgi et al., 2016) which extends the water mass balance in the soil-plant-atmosphere continuum by relating changes in water content directly to water potential and vice-versa. Analogous to the retention curves in soil physics,

the model uses the explicit relationship between water content and potential in plant xylem (pressure-volume curves). The use of plant traits to parameterize such models, as well as their ability to predict measurable features of plant water relations (leaf water potential, sap flow) makes these models attractive from the perspective of both realism and connection to data. Furthermore, such models enable bidirectional water flow through root tissue. Reverse flow from roots to soil is an important process for hydraulic redistribution and tissue dehydration in case of extreme drought (Oliveira et al., 2005; Nadezhdina et al.,

2010; Prieto et al., 2012). A hydraulic model incorporating these advanced mechanisms has recently been included in the Functionally Assembled Terrestrial Ecosystem Simulator (FATES).

FATES is a size and age-structured representation of vegetation dynamics which can be coupled to a land surface model or an Earth system model (Christoffersen et al., 2016). CLM (Lawrence et al., 2019) and the Energy Exascale Earth System Model (E3SM), have both been coupled to FATES and it is used by numerous scientists across the globe to simulate land surface

processes (Koven, 2019).

The new plant hydraulics scheme in FATES was initially developed for specific sites in the tropics, and to our knowledge, porous-media hydraulics models have not been applied extensively, if at all, in high latitude or cold systems. Thus, the physiology of plant hydraulics in winter is generally not well represented in such schemes. The default scheme, for example, stored plant water remains liquid even when temperatures drop below zero and plant gas exchange can continue in the event

of frozen soils. In the cold regions of the world, extreme winters can cause soils to freeze to temperatures well below -20°C. Such cases lead to extremely low water potentials in the soil compared to the plant, while the model assumes that water remains liquid inside the plant. Due to this difference in water potential, soils tend to pull water from vegetation, depleting plant storage pools and incurring large amounts of hydraulic failure mortality in the model.

In this study, we investigate how a model can represent the processes that lead to cold acclimation by implementing a hardening

scheme into the CLM5-FATES-Hydro demographic land surface model. The scheme that represents hardening is configured such that cold-acclimated plants reduce conductances along the soil-plant-atmosphere continuum and the rate for hydraulic failure mortality. We explore additional configurations of the scheme where hardening of plants led to changing pressure





volume curves, and a reduction of the carbon starvation mortality. The modification of these parameters is intended to mimic the physiological changes plants go through when acclimating to cold and the reduction in their metabolic and water transport

capacities. Finally, we assess the sensitivity of the model to the parameters of the hardening scheme (maximum hardiness level and timing of dehardening). We hypothesize that the changes prescribed by the hardening scheme (a) are necessary to model realistic vegetation growth in cold climates, and (b) make it possible to use CLM5-FATES-Hydro to model realistic impacts from frost droughts on vegetation growth and photosynthesis, leading to more reliable projections of how northern ecosystems respond to climate change.

## 2 Methods

### 2.1 The model

CLM5.0-FATES-Hydro stands for the fifth version of the Community Land Model, coupled to the Functionally Assembled Ecosystem Simulator (FATES) with the modified trait based hydraulic scheme (Hydro-TFS). In this context, CLM5.0 represents the biogeophysical aspects of the land surface, including the energy balance, soil hydrology, biophysics and

cryospheric processes, as well as soil biogeochemistry and other land surface types (lakes, urban, glaciers, crops) Lawrence et al. (2019).

### 2.1.1 FATES

FATES (Fisher et al. 2015, Koven et al. 2019) is a cohort-based vegetation demographic model. It has been integrated into CLM5.0 to represent vegetation processes and dynamics wile tracking the vertical (in terms of plant height), spatial (in terms

of successional age) and biological (in terms of plant functional type) heterogeneity of terrestrial ecosystems. When coupled to the CLM5.0 FATES handles all processes related to live vegetation (inclusive of photosynthesis, stomatal conductance, respiration, growth, and plant recruitment and mortality) as well as fire and litter dynamics.

### 2.1.2 Hydro

The 'Hydro' configuration of FATES is a plant hydrodynamic model. Originally developed by Christoffersen et al. (2016), it

follows a continuous porous media approach based on the soil-plant-atmosphere continuum described by Sperry et al. (1998). A single mass balance equation describes the coupled soil-plant system, and the relationship between plant water storage and plant water potential is explicitly described. It consists of a discretization of the soil-plant continuum in a series of water storage compartments with variable heights, volumes, water retention and conducting properties (Fig. 1).

Trees are divided into 4 types of porous media (leaf, stem, transporting and absorbing roots). Each above ground plant medium

and transporting roots consists of a single storage pool, while absorbing roots are divided in vertical compartments by soil layer. The water pools are connected by pathways with defined lengths and conductances. The soil is divided in cylindrical





"shells" around the absorbing roots (the rhizosphere), and hydraulic properties are constant across compartments within a given soil type.

The hydraulics in this scheme are governed by the relationships between content, potential and conductivity of plant tissues. The model requires parameterization of: (1) tissue water content vs. potential (the pressure-volume or PV curve), (2) water potential vs. conductivity of plant tissues and (3) leaf water potential vs. stomatal conductance stress factor. The first two relationships concern every pool within the plant–soil continuum and follow specific equations for plant and soil porous media types. For the soil, we used the formulation of Clapp-Hornberger and Campbell to describe the first two relationships

(Campbell, 1974; Clapp and Hornberger, 1978). For the plant, the first relationship (PV curve) is described by two terms: a) the solute potential (<0 due to the presence of solutes), and b) the pressure potential (>=0 due to cell wall turgor) (Ding et al., 2014; Tyree and Hammel, 1972). The cell wall turgor pressure is the hydrostatic force that pushes the membrane against the cell wall in excess of ambient atmospheric pressure (Fricke, 2017). The equations to calculate the solute and pressure potentials depend on parameters such as the osmotic potential at full turgor (pinot) and the elastic bulk modulus (epsil). The second

relationship (linking plant water potential and conductivity) is calculated using the inverse polynomial of Manzoni et al. (2013) for the xylem vulnerability curve. The fractional loss of total conductance retrieved from this equation is defined at each compartment boundary (Fig. 1) and used to calculate the maximum conductance, $K_{MAX}$. This parameter depends on plant architectural properties and maximum xylem-specific conductivity, and it relates to the maximum attainable flow of water through the xylem. The fraction of maximum stomatal conductance  is used to down-regulate the non-water stressed stomatal

conductance (calculated using Ball-Berry (1987) formulations).

All parameters involved in these relationships are biologically interpretable and measurable plant hydraulic traits. The numerical solution operates every half hour (time-step) and updates water contents and potentials throughout the plant soil continuum.

### 2.1.3 The 2D solver

A crucial component of the porous media hydrodynamics model is the solver used at each time step to find a solution for cohort level water fluxes. The first solver implemented in FATES was a tri-diagonal and 1-dimensional (series of compartments) matrix solver solving the water transport in the plant and rhizosphere soil a layer at a time. Since the solver was not capable of quickly finding a solution during extreme winter conditions, a more efficient two-dimensional (2D) solver using Newton iterations, has been implemented as a replacement. In this solution, the water potentials in plant and rhizosphere

soils are solved together as prognostic variables. Although the new solver improves the quality of the solution and reduces errors, we ran into numerous solver errors when running simulations in Siberia. In very cold soils, extremely low water potentials (-40 000 MPa) lead to extremely high matric water gradients and prevent successful solution of the flow equations (despite increases of the time resolution and number of iterations). In the frozen soil scheme of CLM5, liquid water becomes absent when soils freeze, leading to infinitely small water potentials. Our solution to this problem was twofold:





1) The calculation of supercooling in soils is limited to -10°C since soil temperature can decrease to -40°C in the uppermost

soil layers at extremely cold locations, i.e., East Siberia. This solution yields higher water potential values at all times.

2) We imposed a minimum value on the calculation of soil matric potential from liquid water content of -25 Mpa and a linear

interpolation was used from -25 MPa to -35 Mpa. This implementation allows the solver to efficiently find a solution.

Large rates of water percolation and drainage from upper to deeper soil layers has been reported in earlier modelling

simulations in Siberia (Sato et al., 2010). Our focus is on the physiology of hardening and not on the supply and demand

dynamics of soil moisture, hence we prescribed a depth to bedrock of 1m in all simulations.

## 2.2 The hardening scheme

The temporal dynamics of plant hardiness implemented in this study are based on the work by Rammig et al. (2010). In their

model, the hardiness level ($HD$, in °C) is calculated on a daily basis using three functions: the target hardiness ($TH$, in °C), the

hardening rate ($HR$, in °C day$^{-1}$), and the dehardening rate ($DR$, in °C day$^{-1}$) (Fig. 2). In this context, $HD$ is variable throughout

the year and represents the minimum temperature plants can withstand without incurring injury.

The parameters required for the hardening Scheme by Rammig et al. (2010) are: the minimum hardiness level ($H_{MIN}$); -2°C,

the maximum hardiness level ($H_{MAX}$); -30°C, $HR$; 0.1-1°C day$^{-1}$, $DR$; 0-5°C day$^{-1}$ and several time dependent parameters used

to define the seasons for hardening and dehardening. This scheme was developed to fit the climate of central Sweden

(Farstanäs) and measurements of Norway spruce (Picea Abies). Parameters for the hardening scheme were retrieved from

Kellomaki et al. (1995), Jönsson et al. (2004) and Bigras & Colombo (2013).

In our adaptation of the hardening model, $H_{MAX}$ becomes site- and time- dependent (to function globally and account for

evolution associated to changes in climate), and varies with the 5 year running mean of the minimum 2m daily temperature

($T5$). An exception is made during the first five simulation years. During the first year, $H_{MAX}$ is based on the minimum

temperature of the current year, while from the second to fifth years, it is based on the mean of the minimum temperature of

the preceding years. $H_{MAX}$ can be anywhere between $H_{MIN}$ (-2°C) and a maximum of -70°C (typical in the Arctic and East

Siberia) (Sakai, 1983). The functions used to calculate $TH$ Eq. (1), $HR$ Eq. (2) and $DR$ Eq. (3) were made dependent of $H_{MAX}$

so that vegetation can harden and deharden faster at colder sites where $H_{MAX}$ is lower (Fig. 2 shows the functions for $H_{MAX}$ = -

70°C).

1)   Target hardiness ($TH$)

$$TH = \begin{cases} Hmax & , \; if \; T \leq \frac{Hmax}{1.5} \\ Hmin & , \; if \; T \geq 6 - \frac{Hmax}{6} \\ -sin\left(\pi * \left(0.5 + \frac{T - \frac{Hmax}{1.5}}{-\frac{Hmax}{1.5} + \left(6 - \frac{Hmax}{6}\right)}\right)\right) * \frac{Hmin - Hmax}{2} - \frac{Hmin - Hmax}{2} + Hmin & , \; if \; \frac{Hmax}{1.5} \leq T \leq 6 - \frac{Hmax}{6} \end{cases} \quad (1)$$

2)   Hardening rate ($HR$)



$$HR = \begin{cases} \frac{Hmax-Hmin}{-31.11} + 0.1 & , \, if \; T \leq \frac{Hmax}{2} \\ 0.1 & , \, if \; T \geq 20 \\ \sin\left(\pi * \left(0.5 + \frac{T-\frac{Hmax}{2}}{-\frac{Hmax}{2}+20}\right)\right) * \left(\frac{Hmax-Hmin}{-62.22}\right) + \left(\left(\frac{Hmax-Hmin}{-62.22}\right)+0.1\right), & if \; \frac{Hmax}{2} \leq T \leq 20 \end{cases} \quad (2)$$

3) Dehardening rate ($DR$)

$$DR = \begin{cases} 0 & , \, if \; T \leq 2.5 \\ 5 * \frac{Hmax-Hmin}{-31.11} & , \, if \; T \geq 12.5 \\ (T-2.5) * \left(\frac{Hmax-Hmin}{-62.22}\right) & , \, if \; 2.5 \leq T \leq 12.5 \end{cases} \quad (3)$$

Once a value has been assigned to *TH*, *HR* and *DR*, depending on the daily mean 2m air temperature, the model operates as follows: if *TH* is lower than the hardiness of the previous day (*HDP*), then *HR* is removed from *HDP*. By contrast, if *TH* is higher than *HDP*, *DR* is added to *HDP* Eq. (4).

$$HD = \begin{cases} HDP - HR \, , \; if \; Hdp > TH \\ HDP + DR \, , \; if \; HDP \leq TH \end{cases} \quad (4)$$

Rammig et al. (2010) defines the start of the hardening period as the 210[th] Julian day. We instead use a site specific daylength threshold (*DaylThresh*) depending on the temperature index T5 of the corresponding site Eq. (5 and 6). A site dependent value for *DaylThresh* is essential since vegetation at colder sites must start to harden earlier, and the correlation between photoperiod and temperature is not the same at all locations. Finally, to avoid dehardening in autumn, we keep *HD* fixed during the hardening-only period of the year in cases where *TH* is not below *HDP* Eq. (6).

$$DaylThresh = 42000 + ((-30 - max(-60, min(0, T5)))/15) * 4500 \quad (5)$$

$$HD = \begin{cases} HDP - HR \, , if \; daylength \leq DaylThresh, daylength \; decreases \; and \; HDP > TH \\ HDP \quad\quad , if \; daylength \leq DaylThresh, daylength \; decreases \; and \; HDP \leq TH \end{cases} \quad (6)$$

### 2.3 Physiological impacts of hardening state on plants

Rammig et al. (2010) used the prognostic hardening state to directly modulate the degree to which frost damages plants. In this study, to enhance the mechanistic basis of the simulations, we used the 'hardening' state to simulate the impact of hardening on the hydraulic functioning of the plant, both in terms of the benefits of hardening (i.e. prevention of desiccation) and the physiological costs (reduced ability to conduct water and photosynthesize during the hardening and dehardening phases). We did this by modifying $K_{MAX}$, *g0* & *g1* and the hydraulic failure mortality scalar (see "version zero" simulation in Table 1). We then explore additional configurations of the scheme where hardening of plants led to changing PV curves, and a reduction of the carbon starvation mortality.



### 2.3.1 Maximum conductance between plant compartments ($K_{MAX}$)

In FATES-hydro, $K_{MAX}$ is calculated on the upstream (towards the soil) and downstream (towards the atmosphere) sides of each water compartment connection along the soil-plant continuum. The water potential of the system impacts the direction of the flow and thereby the applicable root radial conductance.

During hardening and water stress, apoplastic ice formation, increased viscosity of protein-bound plant water particles, the increased impermeability of root structures and the inhibition of conducting channels reduce water flow (Dowgert and Steponkus, 1984; Gusta et al., 2005; Smit-Spinks et al., 1984). Therefore, when the hardening scheme is turned on and the hardening level of plants ($HD$) is lower than -3°C (-2°C plus -1°C as margin), the level of hardening is used to exponentially reduce each of the calculated $K_{MAX}$ values along the soil-plant continuum. In "version zero" (Table 1), $K_{MAX}$ was reduced by $10^{\left(\frac{HD+3}{11}\right)}$ Eq. (7).

$$kmax = kmax * 10^{\left(\frac{HD+3}{11}\right)}, \ if \ HD < -3 \tag{7}$$

### 2.3.2 Stomatal conductance (g0 and g1)

The calculation of stomatal conductance in this version of FATES-hydro is based on the Ball-Berry stomatal conductance model. This scheme is typically applied globally, and its parameters and function are not responsive to plant hardening status by default. This means that in winter, when soils are frozen, water loss can still deplete tissue water contents. To prevent desiccation in our scheme, we modify the parameters of the stomatal conductance model as a function of hardening status. The stomatal model has two parameters, the 'minimum' conductance, $g0$ which is the stomatal conductance when photosynthesis reaches zero. In the default setup, $g0$ is fixed at 10 000 $\frac{\mu mol}{m^2 * s}$ for both evergreen needleleaf and deciduous broadleaf trees. The second parameter, $g1$ is the slope of the Ball-Berry stomatal conductance model and is fixed at 8 for all C3 PFTs in the default setup. In our simulations, if $HD$ is lower than -3°C, $HD$ reduces $g0$ and $g1$ in a similar way as the reduction of $K_{MAX}$ Eq. (8 and 9). In the version zero simulation we reduced $g0$ and $g1$ by $10^{\left(\frac{HD+3}{40}\right)}$ Eq. (8 and 9) (Table 1).

$$g0 = g0 * 10^{\left(\frac{HD+3}{40}\right)}, \ if \ HD < -3 \tag{8}$$

$$g1 = g1 * 10^{\left(\frac{HD+3}{40}\right)}, \ if \ HD < -3 \tag{9}$$

### 2.3.3 Hydraulic failure mortality (HFM)

In the default version of FATES-Hydro, HFM (the hydraulic failure mortality) is triggered when the fractional loss of conductivity ($flc$) is larger than a predetermined threshold ($flcThreshold$, set at 0.5). The fractional loss of conductivity is multiplied by a scalar ($MortScalar$, 0.6) that converts the proximal cause of mortality (conductance loss) into a cohort-specific rate of mortality (fraction /year) Eq. (10). As shown in the result section, the default parametrization of HFM used at these cold climate sites, leads to systematic, large and uninterrupted mortality rates throughout the winter on account of desiccation.





Since cold acclimated plants are typically dormant (Chang et al., 2021), and dormant plants are known to have a slower or
interrupted metabolism, we tested a scenario where we reduced HFM with $H_{RATE}$. $H_{RATE}$ is a version of *HD* normalized to the
value of $H_{MAX}$ Eq. (11). $H_{RATE}$ was preferred to *HD* in the reduction of HFM so that if *HD* is equal to $H_{MAX}$, the reduction of
HFM is at a maximum. In the control hardening simulation we reduced HFM by up to 50% at $H_{MAX}$ Eq. (12).

$$HFM = \frac{flc - flcThreshold}{1 - flcThreshold} * MortScalar \qquad (10)$$

$$Hrate = \frac{HD - Hmax}{Hmin - Hmax} \qquad (11)$$

$$HFMortScalar = HFMortScalar * \left( Hrate * \frac{percentage}{100} + \frac{percentage}{100} \right), \; if \; HD < -3 \qquad (12)$$

### 2.3.4 Pressure volume curve

PV curves describe the relationship between total water potential and relative water content in the soil and the plant
compartments. The formulation of the plant compartment PV curves in FATES-Hydro relies on a set of parameters: osmotic
water potential at full turgor/saturation (pinot), bulk elastic modulus (epsil), saturation volumetric water content, residual
volumetric water content, capillary region parameters and relative water content at full turgor (see description of FATES-
Hydro in the model description subsection of the methods). Among these parameters, literature has shown that pinot and epsil
are highly variable, depending on water deficiency. Stressors that induce water deficiency (e.g. drought, cold and frost) trigger
similar responses at the cellular and molecular level. To maintain turgor during stress (Beck et al., 2007) or during hardening
(Valentini et al., 1990), plant organs increase their solute concentration which decreases pinot and they increase the elasticity
of their cell walls which corresponds to a decrease in epsil (Bartlett et al., 2012). While research on desert shrubs has shown
that epsil increased by roughly 10 Mpa during winter (Scholz et al., 2012), literature reveals that pinot easily decreases by 0.5
MPa with hardening and during drought stress (Mart et al., 2016; Valentini et al., 1990).

In this section we describe how we made the PV curves for plant mediums vary throughout the year depending on *HD* (Fig.
3). We lowered the PV curves while daily updating the osmotic potential at full turgor and the elastic bulk modulus Eq. (13
and 14). At an *HD* of -70°C, we lower the default pinot values for leaves (-1.465984), stems (-1.22807) transporting roots (-
1.22807) and absorbing roots (-1.043478) by 0.5 Mpa. The default epsil values (for leaves: 12, stems: 10, absorbing roots: 10
and transporting roots: 8) are increased by 10 MPa at an Hd of -70°C. Unless *HD* gets below -3°C, the default Hydro PV curve
is used, while at its lowest (-70°C), the PV curves are maximally modified. The changes to pinot and epsil modify the shape
of the PV curve so that a given water content is linked to a lower water potential.

$$Pinot = DefaultPinot - \left( 1 - \frac{HD + 70}{67} \right) * 0.5 \qquad (13)$$

$$Epsil = DefaultEpsil + \left( 1 - \frac{HD + 70}{67} \right) * 10 \qquad (14)$$





### 2.3.5 Carbon starvation mortality (CSM)

Similarly to HFM, CSM (the carbon starvation mortality) is triggered when the carbon stored in the leaves is below a target
level, and the fraction of carbon is multiplied by a fixed scalar (set at 0.6 for all plant functional types). CSM is incurred each
winter creating an annual cycle for living biomass of evergreen trees in temperate and boreal regions. In the version zero
simulation, CSM is not reduced (Table 1). In the sensitivity experiments, CSM was reduced following the same method as for
HFM Eq. (12).

### 2.4 Experimental setup

Simulations were carried out using CLM5.0-FATES-Hydro, with fully prognostic state variables for vegetation, litter, and soil.
The atmospheric forcing to drive the model simulations was derived from ERA5-Land data (ERA5L) (ERA5-Land Monthly
Averaged from 1981 to Present). ERA5L provides hourly global high resolution (9km) information on surface variables from
January 1981 to present day, which makes it a valuable dataset for our hardening scheme analysis. In this study, we retrieved
temperature at reference height, wind, humidity, surface pressure, precipitation, downward shortwave radiation and downward
longwave radiation for the entire period. Each simulation was run for 90 years in which the atmospheric forcing from the 30-
year period between 1981 and 2011 was repeated three times. These three periods are depicted as the years 1921 to 2011 for
convenience.

### 2.4.1 Site descriptions

We conducted site-specific simulations for Farstanäs (in Sweden), and near Spasskaya Pad (in Russia). These locations were
selected to verify and illustrate the behavior of the hardening scheme in distinctly different climates.

    1)   Spasskaya Pad is a scientific research station in the taiga near Yakutsk, Russia (62°N and 129°E), the coldest large
city in the world, with an annual average temperature of roughly -9°C. Spasskaya Pad has never recorded a
temperature above freezing between the 10[th] of November and the 14[th] of March and the average winter temperature
is below -20°C. However, the warm summers (with a July average and highest daily mean temperatures of ~20°C and
~25°C respectively) place Spasskaya Pad far south from the tree line. The total yearly precipitation is around 280
mm, and the snow depth typically reaches 40 cm. Spasskaya Pad is in the center of Yakutia, the largest republic of
the Russian Federation, which is mostly covered by boreal vegetation (74%) (Isaev et al., 2010). The forests are
mainly composed of Larch (deciduous needleleaf), patches of Scots pine (evergreen needleleaf) on sandy soil
(Sugimoto et al., 2002), dwarf Siberian pine (*Pinus pumila*) and to a lesser extent Siberian spruce (*Picea omorika*),
and small stands of birch (*Betula*), fir (*Abies*) and aspen (*Populus*).

    2)   Farstanäs (59°N and 17°E) is slightly south of Stockholm and presents a cold temperate climate. The mean yearly
temperature is around 5.5°C and the total yearly precipitation is around 800 mm. The vegetation at Farstanäs is a
mixed forest with, among others, spruce, pine, beech, oak, elm, ash, and maple.





### 2.4.2 Plant functional type

To remove species competition and better understand the impacts of hardening on vegetation growth, we performed each simulation with only one plant functional type. We tested the scheme on two plant functional types: extratropical evergreen needleleaf trees and cold-deciduous broadleaf trees. Although evergreen needleleaf trees are not the dominant plant type at Spasskaya Pad (Petrov et al., 2011; Tatarinova et al., 2017; Hamada et al., 2004), we selected this location since it is one of the most extreme (cold and dry) climates where pine trees still exist. Therefore, we expect that the benefit from introducing a

cold hardening scheme may be particularly apparent at this location. Results from broadleaf deciduous simulations are included in the supplemental (Fig. S3). Note that soil conditions, including matric potential, are similar in deciduous and evergreen simulations (Fig. 4).

### 2.4.3 Main simulations

In the first part of our results we compare the implementation of the new hardening scheme (version zero) into CLM5-FATES-

Hydro to the default version of the model. In the version zero of the hardening scheme, cold-acclimated plants reduce I) the maximum conductance between plant tissues, and between roots and soil ($K_{MAX}$), II) both the intercept and the slope of the stomatal conductance model, and III) the rate for hydraulic failure mortality (HFM) (Table 1). For the version zero simulation, the values for the reductions of $K_{MAX}$, $g0$ and $g1$, $DR$ and $H_{MAX}$ were selected based on preliminary testing to minimize winter water losses and to maximize vegetation biomass at Spasskaya Pad and Farstanäs. In the version zero hardening simulation,

dehardening ($DR$) is described by a linear function that increases from 2.5°C to 12.5°C for evergreen trees Eq. (3). For reasons discussed further down, the PV modifications were not selected in the version zero.

### 2.4.4 Sensitivity experiments

To test the sensitivity of vegetation growth to the amplitude at which $K_{MAX}$, $g0$ & $g1$ and HFM are modified by hardening, we tested individual modifications for each of these parameters (Table 2). Compared to the magnitude of the reductions selected

in the version zero simulation, we selected both stronger and weaker reductions for the sensitivity experiments. Additional simulations were run to assess the impact of g0 and g1 independently from each other. We further tested extra implications which cold acclimation may have on processes in FATES-Hydro, such as the modification of PV curves and the reduction of CSM with hardening (Table 2). For CSM, the reductions with hardiness follow the same method as for HFM Eq. (12). In contrast to the other CSM experiments, the one called "50% all year" does not depend on the hardiness level but directly on

$H_{MAX}$ instead. Two additional sensitivity experiments were performed on the temperature range of the dehardening rate ($DR$) and on the maximum hardiness level ($H_{MAX}$) – parameters involved in the calculation of the hardiness level (Table 2). To evaluate the sensitivity of the hardening scheme to $DR$, we ran an experiment where the $DR$ function increases between 0° and 10°C and an experiment where $DR$ increases between 5°C and 15°C (Table 2). While $H_{MAX}$ is defined as T5 minus 10°C in the version zero simulation, we ran experiments where $H_{MAX}$ was defined by T5 minus 5°C and T5 minus 15°C.





## 3 Results

### 3.1 Hardening to survive in the arctic

At full spin-up, the default CLM5.0-FATES-Hydro model, without hardening, yielded evergreen tree biomass of ~40 MgC ha$^{-1}$ at Farstanäs and only ~0.2 MgC ha$^{-1}$ at Spasskaya Pad (Fig. 4a and b). After inclusion of the hardening scheme, larger biomass levels are simulated in Spasskaya Pad (~4.5 MgC ha$^{-1}$), and similar levels at Farstanäs (~42 MgC ha$^{-1}$).

In Spasskaya Pad, the temperature of the top soil layers drops below -20°C each winter (Fig. 5d). The resulting low liquid water content of the soil leads to such low matric potentials that the default model systematically simulates a release of water from the plant to the frozen soil (Fig. 6d). The root water release is strongest in autumn when soils start to freeze, but it can continue deep into the winter as long as plants still have stored water and the soil matric potential decreases further. If the top soil temperature remains higher than -25 °C, our results show that winter dehydration typically leads to plant matric potentials higher than -15 MPa (Fig. 7b). During extreme years with top soil temperatures below -25 °C, the matric potential in plant tissues can sometimes get as low as -28 MPa. The repetition of long lasting low soil matric potentials one winter after the other, resulted in large HFM rates (> 0.55 % individuals year$^{-1}$) (Fig. 8b and c). The sum of HFM, CSM and other minor mortalities, outweighs the summer productivity of needleleaf trees in Spasskaya Pad (Figs. 4b and S4b).

When the hardening model is employed in FATES-hydro, low winter temperatures at Spasskaya Pad mean that *HD* quickly reaches the fixed limit of -70°C (Fig. 10). This results in a strong reduction of I) $K_{MAX}$, II) *g0* & *g1* and III) HFM (See methods above and sensitivity sections below). The reduction of $K_{MAX}$ between tissues, and especially between absorbing roots and the first rhizosphere, greatly reduces the amplitude of reverse water flow through the roots when soils are frozen (Fig. 5c and d). The temporal dynamics of the hardening model allow for cold-induced damage. For example, if temperatures drop abruptly below freezing in autumn, plants won't have acclimated yet and the amplitude of the reverse water flux will therefore be similar to the default model. However, as plants start acclimating, the resistance to water flux increases and root water exudation is inhibited by the hardening scheme. The second implication of hardening in our hydraulically-adapted hardening scheme plants is the reduction of *g0* & *g1*. This reduces transpiration in spring and enables leaves to maintain their water potential while plants are still cold acclimated. The reduction of the conductances ($K_{MAX,}$ *g0* and *g1*) results in larger amounts of stored plant water (Fig. 6f). Levels of stored plant water are ~2.4 Kg KgC$^{-1}$ during summer, decrease to around ~1.75 Kg KgC$^{-1}$ during warmer winters and reach as low as ~1.1 Kg KgC$^{-1}$ during extremely cold winters. In the default model, stored water drops to ~0.6 Kg KgC$^{-1}$ without major variation between years. By keeping larger amounts of water in hardened plants, the fractional loss of total conductance (flc) remains lower than in default simulations. The hardening scheme greatly reduces hydraulic failure mortality (HFM) at Spasskaya Pad (Fig 8b), since HFM is a function of flc, and an additional direct reduction of HFM was applied during cold acclimation (Fig S12b). The contribution of the changes to $K_{MAX}$, *g0* and *g1*, and the reduction of HFM, can be seen by comparing Fig. 8b and Fig. S12. The changes imposed through hardening favors vegetation growth in northern regions while the default model simulates almost nonexistent and declining vegetation (Fig. 4b).





At Farstanäs, temperatures in the top soil layers usually remain between 0 and -5°C during winter (Fig. 5c). This means that the water potential in plant compartments rarely drops below -3 MPa during typical winters. During cold winters, plant water potentials remain above -6 MPa in both the default and the hardening versions of the model (Fig. 7a). Therefore, the rate of
HFM is lower at Farstanäs than at Spasskaya Pad, and episodes of mortality are shorter (Fig. 8a). In the simulations with hardening, plants have higher water potentials, reflective of the lower rates of water loss from winter root water exudation and stomatal transpiration. The main reason why the changes induced by hardening are small in Farstanäs – compared to Spasskaya Pad – is that changes are proportional to $HD$, which does not decrease much at Farstanäs (Fig. 10), and therefore the reductions applied to the conductance and mortality are smaller.

The simulations at Spasskaya Pad and Farstanäs, both feature years with notably large drops in living biomass (Fig. 4a and b). These are related to low soil water potentials during winter (Fig. 7a and b). The unusually low soil matric potentials in these years contribute to an increase in dehydration and lead to higher mortality rates (Fig. 8). Our results show that a stronger reduction of $K_{MAX}$ with hardening, a lower $H_{MAX}$, or a stronger reduction in HFM led to larger survival rates. At Spasskaya Pad, none of the sensitivity simulations prevented the strong mortality rates during extreme years (Fig. 11). While an even stronger
reduction of $K_{MAX}$ and a later dehardening slightly helped survival, all simulations went through approximatively 50% less mortality during the dry years. Figs. S2b and 4b illustrate that during these extreme years, the total precipitation was low and the snow layer was thin. The water stored in plants and the matric potential of plants recovered only in the middle of the following summer (Fig. 6f, 7b).

## 3.2 Sensitivity experiments

### 3.2.1 Dehardening rate (DR)

At both sites (Farstanäs and Spasskaya Pad) the earlier dehardening starts, the better it is for vegetation growth (Figs. 11 and 12). There seem to be limited benefits in delaying hardening in autumn. At Spasskaya Pad, the modification of dehardening has little influence in the middle of the winter because of the extremely low hardiness levels ($HD$) (Fig. 10b). However, the "early dehardening simulation", in which hardening decreases later and increases earlier ($DR$ between 0° and 10°C) shows
root water efflux at the beginning and at the end of the winter season (Fig. S4d). This results in lower stored plant water and thereby higher hydraulic failure mortality (Figs. S4f and S5b). Interestingly, this results in a trade-off with carbon starvation mortality, since plant metabolism is activated earlier, reducing CSM (Fig. 9) and increasing gross primary productivity (Fig. S6).

At Farstanäs, soils are rarely frozen in autumn and spring, and when they freeze it is never as much as in Spasskaya Pad (Fig.
5). This, combined with the shorter cold seasons and the higher hardiness levels during winter, results in a reduced occurrence of hydraulic failure mortality compared to Spasskaya Pad (Fig. S5). The 30-year atmospheric forcing period is too short and variable to show the trade-off between hydraulic failure mortality and carbon starvation mortality or even gross primary productivity.



### 3.2.2 Maximum hardiness level (H$_{MAX}$)

At Spasskaya Pad, the $H_{MAX}$ sensitivity simulations (where $H_{MAX}$ is predicted by T5 minus variable Celsius degrees, see Table 1) are close to the hardening limit of -70°C (Fig. 10b), which means that they yield similar results (Fig. 11).

At Farstanäs, lowering $H_{MAX}$ had large impacts on the total biomass (Fig. 12) because it increases survival during years with low soil water potentials and strong mid-winter root water release (Fig. S5). Conversely, it reduces productivity during years with higher soil water potentials (Figs. S4 and 12).

In our model, a reduction in $H_{MAX}$ lowers conductivity, introducing a cost to plants in the form of decreased transpiration and photosynthesis (Fig. S4). Our results show the increase in CSM and decrease in HFM due to the lowering of Hmax at Spasskaya Pad (Fig. S7).

### 3.2.3 Pressure volume (PV) curve

PV curves were produced by making two of their parameters dependent on the hardening status $HD$ (see method section). Our
implementation results in a shift in the PV curves when plants cold acclimate, and associates a lower matric potential with a given volumetric water content when they are hardened (Fig. 3). By decreasing the water potential in the plant, we effectively decrease the water gradient between freezing surface soil layers (generally surface layers). However, we simultaneously increase the water gradient between deep roots and unfrozen, usually deeper, soil layers. Since plants tend to establish equilibrium with the soil, stronger water uptake in deep roots decelerates the decrease in plant water potential.

Comparing Farstanäs and Spasskaya Pad reveals that the level of reduction of $K_{MAX}$ is crucial to the effective functioning of the dynamic PV curves. At Farstanäs (Fig. S8a), deep layers of the soil rarely freeze, and the weak reduction of $K_{MAX}$ enables water uptake, thereby offsetting the direct effect of $HD$ on the PV curves. In Spasskaya Pad (Fig. S8b), the strong reduction in $K_{MAX}$ and the resulting prevention of water fluxes between roots and soil enabled lower water potential in the plant compartments. This led to slightly larger HFM rates and lower vegetation biomass (Fig. 11). We note that the higher leaf water
potential in the dynamic PV simulation in Farstanäs in 2003 (Fig. S8a), is only due to differences in snow depths, and changes to soil temperatures resulting from different biomasses between both PV sensitivity simulations.

### 3.2.4 Maximum conductance between plant compartments ($K_{MAX}$)

In this section, we show how sensitive evergreen trees are to the rate at which $HD$ reduces $K_{MAX}$ Eq. (7). In Farstanäs, weaker reductions of $K_{MAX}$ were favourable to vegetation productivity during "mild" winters, while stronger reductions became
beneficial during "cold" winters (Fig. 12). During a cold year (e.g. first year of Fig. S9 a, c and e), the simulation with strong reduction in $K_{MAX}$ allows plants to lose less water to root exudation. Since trees hold on to more water, a larger exchange of water and carbon for photosynthesis is possible during early spring. By contrast, during a "mild" year (e.g. second year of Fig. S9 a, c and e), the strong reduction in $K_{MAX}$ does not provide an advantage for the plant since soil water potentials remains high





and strong root water exudation is absent even in the default model run (Fig. 12). Again, our model captures here the costs and
benefits of hardening. Overall, it seems like the medium reduction of $K_{MAX}$ yields the highest biomass in Farstanäs (Fig. 12).
At Spasskaya Pad, large reductions in $K_{MAX}$ are always necessary to allow persistence of living biomass in our simulations
(Fig. 11). Moreover, the simulations with the largest reductions of $K_{MAX}$ have the highest vegetation biomass.

### 3.2.5 Minimum stomatal conductance (g0 and g1)

Lowering g0 & g1 with hardening greatly reduces transpiration during winter (Fig. S10). This allows leaf water potentials
(Fig. S11) to be maintained at values that do not trigger mortality. Reducing g0 and g1 also appears to lead to higher leaf water
potentials during summer, especially at Farstanäs, potentially due to lower rates of transpiration resulting in larger water
availability in the soil during summer (Fig S10e and f). Overall, the reduction of g0 and g1 has a positive impact on the living
biomass at both sites (Fig. 11 and 12).

Reducing only g0 leads to water loss when light levels increase the stomatal response, and reducing only g1 still allows
substantial transpiration to occur when stomata are shut. Both need to be reduced to avoid a loss of internal plant water stores
during wintertime (Fig. S11 a and b).

### 3.2.6 Hydraulic failure mortality

If the rate scalar of HFM is not modified by the hardening scheme, high mortality rates are simulated for extended periods
during winter at Spasskaya Pad (green line in Fig. S12b). When the rate of HFM is reduced by 50% and 100% at $H_{MAX}$ (brown
and dark green lines respectively in Fig. S12), this results in a large increase in vegetation biomass (Fig. 11). A 50% reduction
of HFM leads to almost a doubling of the biomass, and a 100% reduction (not realistic but included as an edge case) to almost
a quadrupling.

On the other hand, the larger amounts of living biomass generated by the reduction of mortality, imply that larger amounts of
water are transpired and there is more competition for soil liquid water. During extreme years, it appears that vegetation in
simulations with reduced HFM, suffers from larger rates of CSM, which does not completely disappear during the following
summer (Fig. S12d). In addition, while there was no HFM in the unchanged HFM simulation in spring, plants in reduced HFM
simulations incur small amounts of HFM until later into the following summer (Fig. S12d).

### 3.2.7 Carbon starvation mortality

Implications of changing the rate of CSM are minor compared to HFM, although both scaling factors are identical (0.6 is the
maximum mortality rate). HFM is parametrized in such a way that it quickly reaches 0.6 during strong water stress, while
CSM remains at ~0.3, even during the more extreme years (Fig.S13a and b). The biomass response of a reduction in CSM is
similar at both sites: vegetation thrives best with reduced CSM (Fig. 11 and 12).



## 4 Discussion

Much of the recent development of vegetation dynamics in land surface models has focused on representing advanced plant
hydrodynamics (Christoffersen et al., 2016; Kennedy et al., 2019; Sperry et al., 2017; Xu et al., 2016). Usage of schemes that
simulate the internal dynamics of plant moisture, however, have rarely been tested in cold systems – if at all, to our knowledge.
By integrating a model of plant hardening with a porous media approach to plant hydrodynamics, we integrate a set of
mechanisms that are both necessary for plants to avoid winter desiccation and capture the costs (in terms of reduced growth in
spring) and benefits (in terms of reduced mortality rates) of winter 'hardening'. Our analyses further highlight trade-offs
between avoidance of 'frost drought' mortality via hardening and avoidance of carbon starvation mortality via early season
photosynthesis.

The default version of the CLM5-FATES-Hydro model used here, allows water to freeze in soils but not in plants. It does not
include a mechanism to prevent liquid plant water to flow from plants to soils when freezing strongly reduces soil water
potential. We show that this results in a depletion of water in plant compartments and triggers large amounts of hydraulic
failure and carbon starvation mortalities.

The hydraulically-mediated hardening scheme we propose in this paper, consists of three modifications. The first being a
reduction in the hydraulic conductivities of plant tissues which inhibits or prevents water loss during freezing depending on
the atmospheric temperatures and amount of hardening by the plants (Gusta et al., 2005; Smit-Spinks et al., 1984; Steponkus,
1984). While the benefit of hardening is the reduced hydraulic failure mortality during freezing soil events, its cost is a
temporary reduction of photosynthesis due to reduced transpiration. The sensitivity experiments on the timing of dehardening
($DR$) and on the maximum hardiness level ($H_{MAX}$) of this study, highlight the considerable cost of hardening in cold sites which
incur frequent hydraulic failure mortality (Fig. 9, S6, S7). Previous field-based research has described the cost of hardening
by showing that (1) cold acclimation causes a suppression of the rate of $CO_2$ uptake (Krivosheeva et al., 1996), (2) low
temperatures lead to the inhibition of sucrose synthesis and photosynthesis (Savitch et al., 2002; Stitt and Hurry, 2002), and
(3) photosynthesis stops when needles freeze (Havranek and Tranquillini, 1995). In addition, growth cessation, dormancy and
cold acclimation are closely related to each other (Chang et al., 2021).

The second modification is to the stomatal model of Ball and Berry (1987). Reducing g0 & g1 led to a slower decrease of leaf
water potentials during late winter and spring, resulting in higher vegetation survival rates. We argue that the accompanying
reduction in transpiration makes the model more realistic. Indeed, while photosynthesis is slightly influenced by cold in the
default model, temperature is not taken into account in the calculation of transpiration, although literature has shown that g0
& g1 is lower in cold acclimated plants (Christersson, 1972; Duursma et al., 2019; James et al., 2008). A partial but not full
reduction of g0 and g1 allows for some transpiration, and thereby maintaining the capacity of the advanced hydraulic model
to represent winter droughts. In contrast to summer droughts, typically caused by the absence of rainfall in summer, winter
droughts or frost droughts are caused by the unavailability of water in frozen soils, preventing the replacement of transpired
water during warm winter days.



The third modification that comprises our hardening scheme is the reduction in the rate of HFM. While the default formulation of HFM might be realistic when simulating summer droughts, it is not adapted to boreal and arctic regions where winter droughts would cause extended periods of mortality (i.e. lasting until snow melt in spring). At cold sites, the default CLM5-FATES-Hydro simulates a frost drought each winter due to root water exudation which systematically results in maximum

HFM rates from late autumn until spring snow melt when soils finally thaw. The current parameterization of HFM in cold regions is such that summer productivity cannot balance the high winter mortality. In reality, high latitude vegetation is dormant in winter and the metabolism of plants is reduced or completely interrupted (Volaire, 2018). When dormant, plants reduce or stop meristem activity to make it insensitive to growth and promote signals in order to enhance survival during seasons with life threatening environmental conditions (Volaire, 2018). To quote Volaire (2018), "knowing when not to grow

does not confer drought resistance but may well enhance drought survival". Our results show that the reduction of HFM with hardening can lead to large increases of biomass in areas with cold winters. The large amounts of biomass, however, appear to be a limiting factor during dry years, when plants must compete for water. The larger amount of biomass caused by reduced HFM rates resulted in higher CSM rates, highlighting another trade-off emerging from the hardening scheme.

To tolerate freezing, plants undergo a set of physiological changes. The capacity to efficiently cold acclimate and survive frost

depends on a plants' genome and the presence of performant cold tolerance traits. Therefore, in reality different species, may exhibit different maximum hardiness levels ($H_{MAX}$), hardening rates ($HR$), dehardening rates ($DR$) and temperature ranges where hardening and dehardening occur (Mabaso et al., 2019; Oberschelp et al., 2020). In terrestrial biosphere models, species are aggregated in groups based on common functional characteristics criteria. Therefore, it is likely that the measured maximum freezing tolerance of some species that are within the same plant functional type category in the model, is variable

in reality. (Sakai, 1983). A PFTs maximum freezing tolerance (as well as most parameters of terrestrial biosphere models) is a rough approximation of what has been measured for species belonging to that PFT. Common garden are generally used to identify the genetic variations among populations in their ability to cope with stresses (Bansal et al., 2016; de Villemereuil et al., 2016). As the expression of physiological/morphological traits associated with stress tolerance is also dependent on the environmental conditions of a common garden, several gardens are required to quantify the relative influence of environment

and genetics on the expression of stress tolerance traits (Greer and Warrington, 1982). Earlier studies used a $H_{MAX}$ of -30°C and a $DR$ initiated at 5°C to model the hardiness of *Norway spruce* in Farstanäs (Bigras and Colombo, 2013; Jönsson et al., 2004; Rammig et al., 2010). However, *Pinus sylvestris* seedlings in Finland start dehardening already at temperatures as low as 3°C (Repo and Pelkonen, 1986). Our hydraulically-mediated hardening scheme captures both the costs and benefits of hardening, which in simulations with dynamic vegetation would likely lead to different competitive outcomes for plants with

alternative hardening thresholds. Finer discretization of plant strategies along the axis of cold tolerance would provide an interesting extension to this study.

The physiological changes induced by cold acclimation result in modifications of the osmotic potential at full turgor and the bulk elastic modulus (Mart et al., 2016; Scholz et al., 2012; Valentini et al., 1990), i.e. two parameters that intervene in the calculation of the plant PV curves. In the changing PV curve simulation (see methods for more details), lower winter root





water exudation and transpiration fluxes were simulated. If $K_{MAX}$ is not low enough, larger root water uptake (especially in deeper layers that remain unfrozen longer in autumn) balances the decrease in plant water potential. While the PV curve modification should be a more realistic approach to model plant hydraulics, it also results in larger HFM and lower biomass. Our study illustrated the potential for a trade-off between the avoidance of HFM in the spring and the avoidance of CSM triggered by low photosynthesis rates resulting from a long hardening season. This trade-off mirrors the one originally proposed

by McDowell et al. (2008) for summer droughts (whereby stomatal closure avoids HFM but predicates CSM). Mature trees store large amounts of mobile carbon, which decrease under water stress, but evidence to support this hypothesis is lacking (Hartmann, 2015; Sala, 2009). While HFM can be assessed by the percent loss of total conductance, CSM is much harder to infer. The limited understanding of the roles of non-structural carbohydrates suggests a link between CSM and HFM as sugars not only are a source of energy, but also regulate osmotic pressure and embolism repair following drought (McDowell &

Sevanto, 2010). Our results show that changes in CSM result in relatively small changes of biomass compared to HFM reductions. This is mainly due to the fact that droughts typically cause larger HFM than CSM episodes.

## 5 Conclusions

In this study, we propose a hardening scheme adapted for use within the context of plant hydrodynamic simulations, which can simulate the physiological costs and benefits of plant cold acclimation in terms of water movement and gas exchange. Its

impact on plant hydraulics and vegetation mortality and growth appears to be a promising improvement for the modelling of vegetation growth in cold environments. We present here one parameterization of the hardening scheme, show how it performs at two sites with contrasting winter weather, and investigate the response of the scheme to variations to its key parameters. Although the understanding about cold acclimation processes is expanding at an accelerating rate, there are still large knowledge gaps. For example, the range of processes triggered by cold acclimation are poorly understood and we lack

measurements to define the exact amplitude at which they are disturbed (Arora and Rowland, 2011; Chang et al., 2021; Shi et al., 2018; Shin et al., 2015). In addition, quantifying and generalizing hardiness levels and rates inside the scheme itself are in some respects broad approximations which remain to be optimized further. Future developments might, for example, consider a larger influence of photoperiod and the inclusion of plant phenological states in the calculation of the hardiness level. In this study, we primarily aim at discussing the role and impacts of major parameters and potential impacts of cold acclimation as a

framework for further implementation in dynamic vegetation models with advanced plant hydraulics. We show that this framework 1) leads to more realistic vegetation biomass productivity at temperate and boreal sites, 2) influences winter root water release and mortality rates by lowering plant conductance, and 3) that hardening comes at a cost for photosynthesis (trade-off of hardening emerges from our scheme).

Recent observations of increasing vegetation mortality appear to be a result of climate change, in particular the increase in

intensity and frequency of droughts caused by extreme weather conditions (Allen et al., 2010). This highlights the urgency to improve our understanding of plant survival and mortality mechanisms. To date, there are large gaps in our knowledge on



plant hydraulics and their link to mortality rates. In this paper, we hope to provide new insights into modelling of plant hydraulics and their link to cold acclimation.

Future research is needed to better assess the implications of cold acclimation on plant hydraulics, especially conductivity.
Understanding these processes in hampered by the logistical and technical difficulties involved in the observation of cold systems. Our work lays the foundation to use a hardening scheme to regulate frost damage and to study the link between different types of mortalities in terrestrial biosphere models in the Arctic region. The inclusion of cold hardiness is essential to model realistic plant hydraulics and vegetation dynamics within cold climates.

**Data availability**

The modelling data that supports the findings of this study is available at https://doi.org/10.11582/2022.00028. The code of CTSM5.0 can be found at https://zenodo.org/badge/latestdoi/493596086, and the code of FATES at https://zenodo.org/badge/latestdoi/493596262.

**Acknowledgements**

We gratefully acknowledge the support of the Research Council of Norway for the WINTERPROOF (project no. 274711), the
Swedish Research Council under registration no. 2017-05268, the EMERALD (project no. 294948), and the Center for Biogeochemistry in the Anthropocene at the Faculty of Mathematics and Natural Sciences at UiO. We would also like to thank the ECMWF for the ERA5L reanalysis product. Finally, we would like to acknowledge our colleagues at NCAR and Berkeley lab for their precious help during model setup and manuscript preparation. RF acknowledges the support of the US Dept of Energy 'Next Generation Ecosystem Experiment in the Tropics' project, as the EUH2020 4C project. The simulations were
performed on the FRAM supercomputer operated by Sigma2 (project number NN2806K).

**Author Contribution**

MSAL, HT, KJA, FS, RAF and FJWP designed the work; MSAL performed experiments and data analyses; MSAL drafted the manuscript; HT, KSA, FS, JWB, RAF, YF, JD and FJWP revised the manuscript. All authors approved the final version of the manuscript for publication. The authors declare that they have no conflict of interest.

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

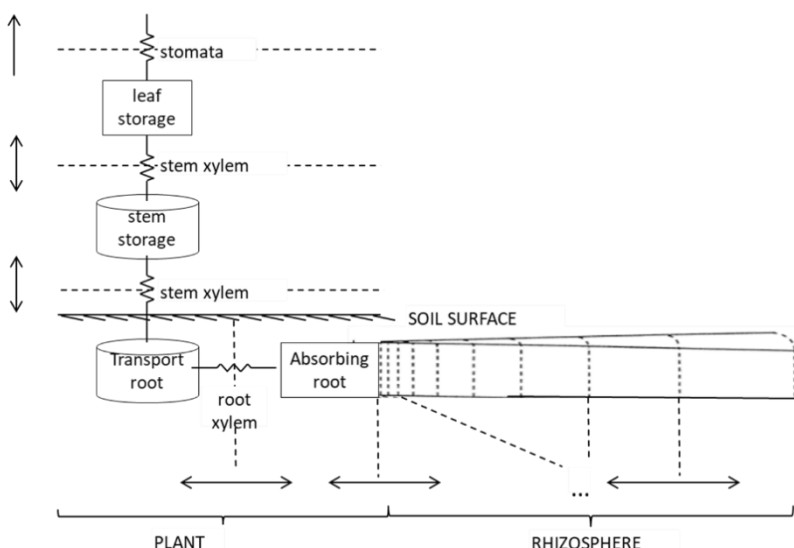

**Figure 1: Structure of the FATES-Hydro model (Xu et al., 2020).**



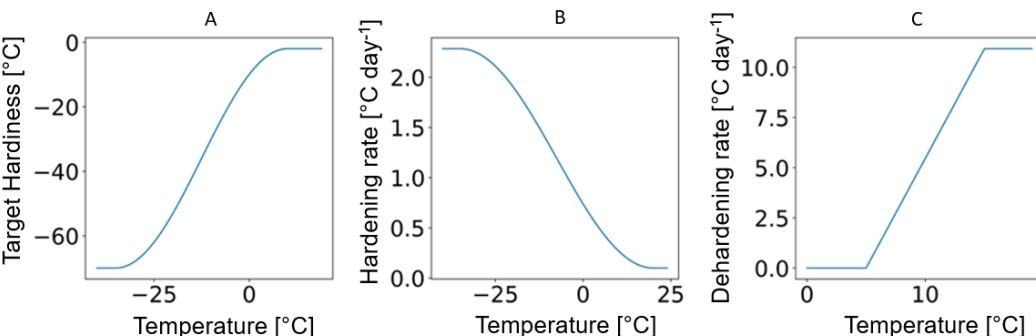

Figure 2: Example of (a) the target hardiness (TH), (b) the hardening rate (HR, in °C day−1), and (c) the dehardening rate (DR) functions (in °C day−1) in relation to the ambient mean temperature corresponding to a site and a plant functional type with a maximum hardiness level (H$_{MAX}$) of -70°C.

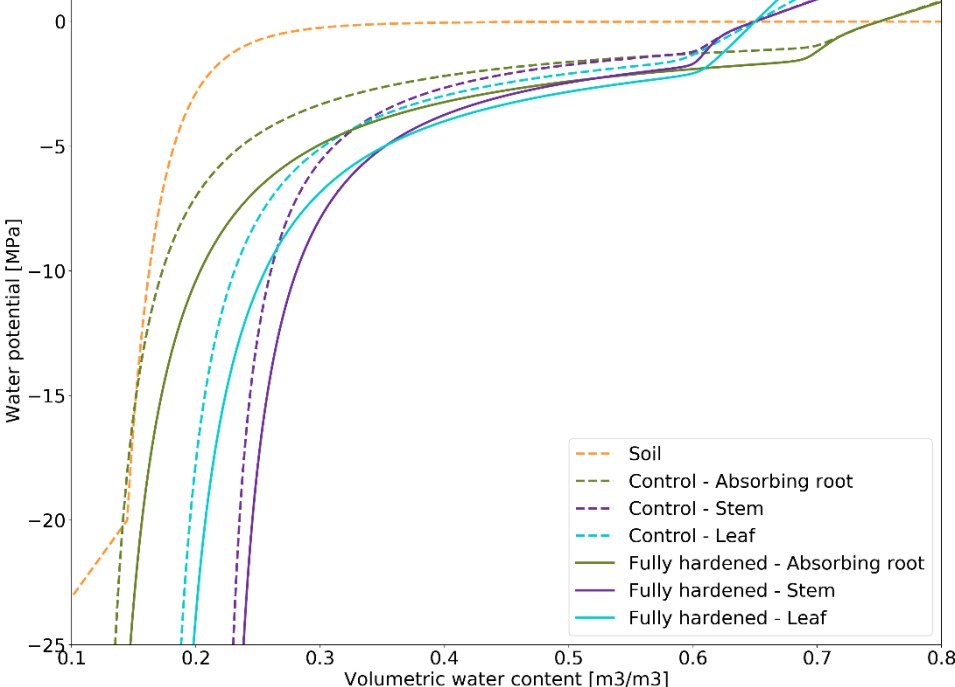

Figure 3: Pressure volume curves for soil (orange) and plant compartments. The PV curves used by the default CLM5-FATES-Hydro and non-hardened plants are the dashed lines, while the PV curves used by plants acclimated to -70°C are the full lines.



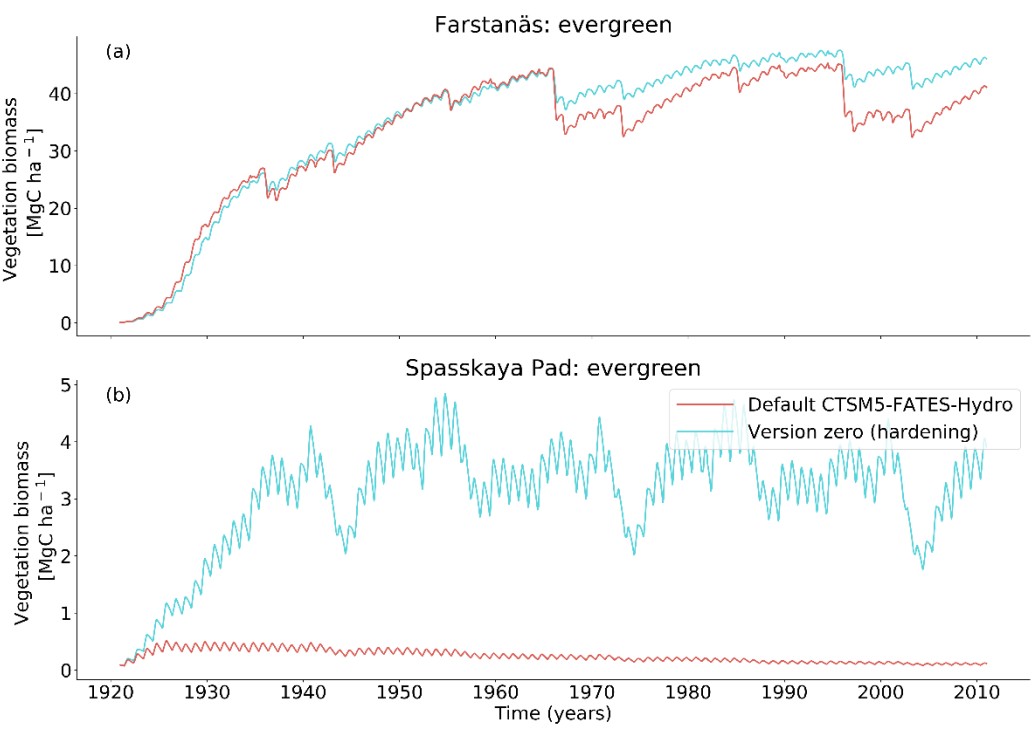

**Figure 4: Living biomass for needleleaf evergreen trees at the sites of a) Farstanäs, and b) Spasskaya Pad, during the period 1921-2011 (atmospheric forcing: 3\*[1981-2011]). The default simulation is shown in red, and the hardening simulation is shown in blue.**

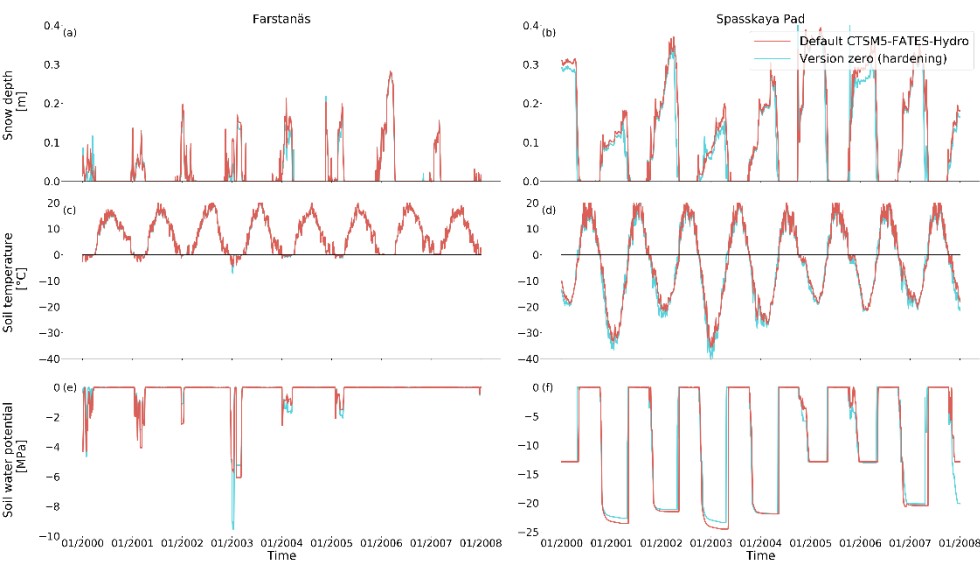

**Figure 5: Soil conditions for needleleaf evergreen trees at the sites of: Left) Farstanäs, and Right) Spasskaya Pad, during the period 2000-2008. Top: transpiration, middle: root water uptake, and bottom: stored plant water. The default simulation is shown in red, and the hardening simulation is shown in blue.**





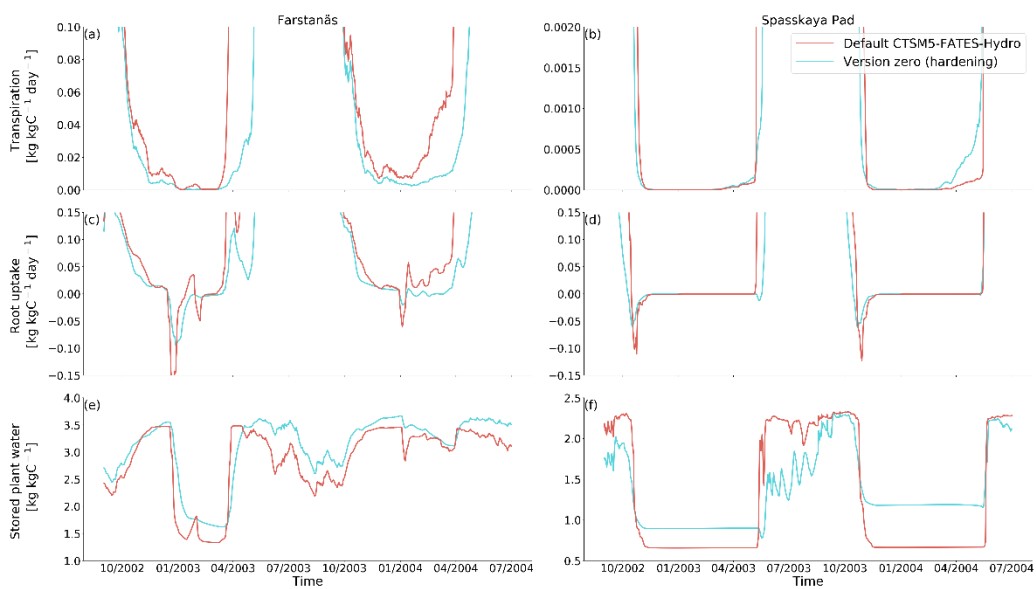

**Figure 6: Plant water fluxes for needleleaf evergreen trees at the sites of: Left) Farstanäs, and Right) Spasskaya Pad, during the period 2002/09-2004/07. Top: transpiration, middle: root water uptake, and bottom: stored plant water. The default simulation is shown in red, and the hardening simulation is shown in blue.**

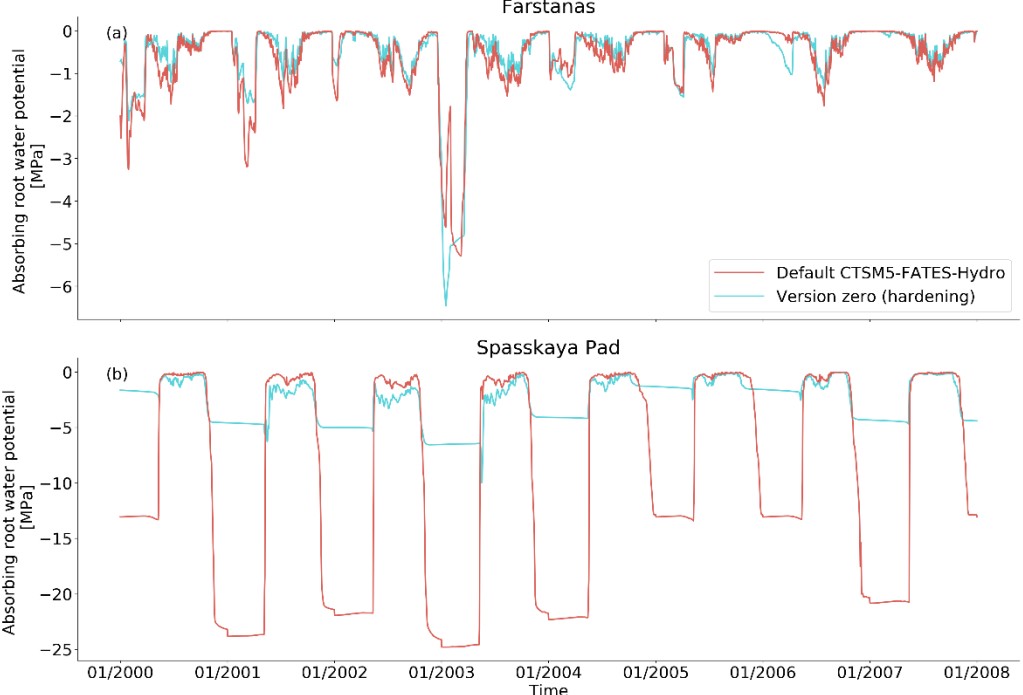

**Figure 7: Absorbing root water potential for needleleaf evergreen trees at the sites of a) Farstanäs, and b) Spasskaya Pad, during the period 2000-2008. The default simulation is shown in red, and the hardening simulation is shown in blue.**

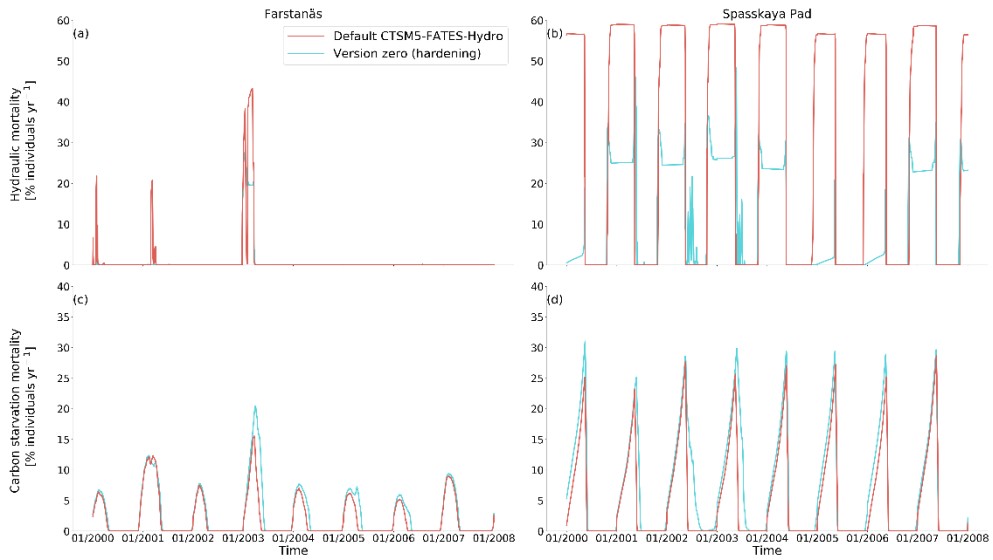

**Figure 8: Mortality rates for evergreen needleleaf trees at the sites of: Left) Farstanäs, and Right) Spasskaya Pad, during the period 2000-2008. Top: hydraulic failure mortality, and bottom: carbon starvation mortality. The default simulation is shown in red, and the hardening simulation is shown in blue.**

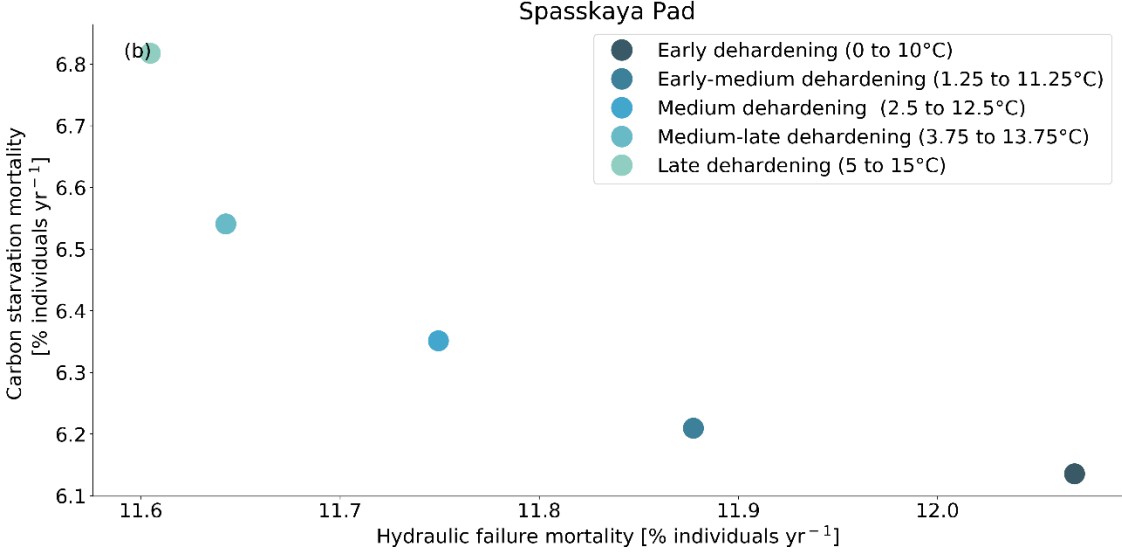

**Figure 9: Trade-off between hydraulic failure mortality and carbon starvation mortality for evergreen needleleaf trees at Spasskaya Pad for 5 dehardening sensitivity experiments. The mortality rates are averaged over the 30 year period 1981 to 2011.**



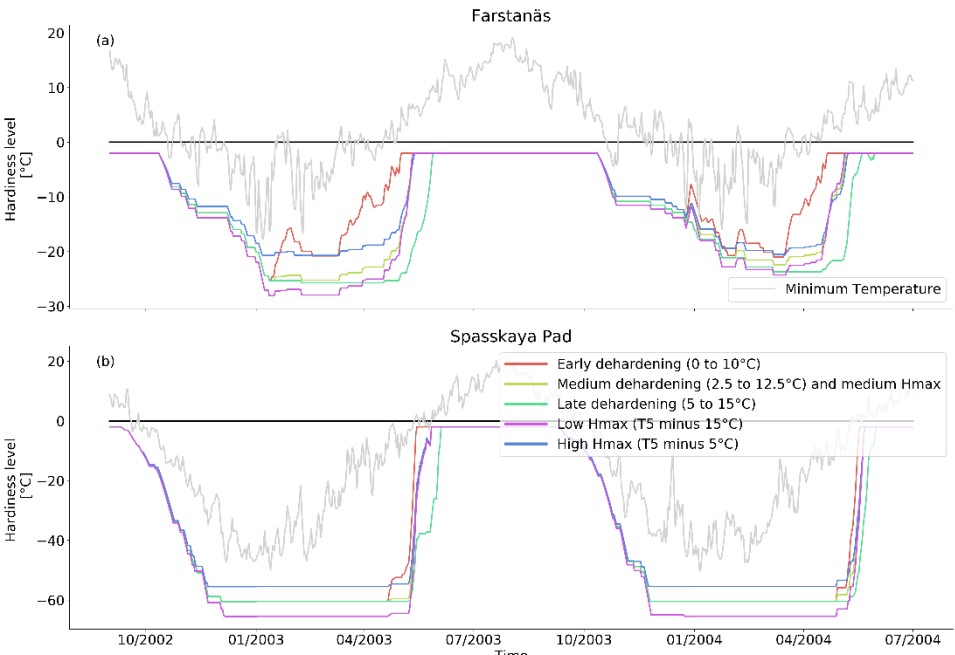

**Figure 10: Hardiness level from dehardening and maximum hardiness level sensitivity analysis simulations for needleleaf evergreen trees at the sites of: a) Farstanäs, and b) Spasskaya Pad, during the period 2002/09-2004/07. The grey line corresponds to the**
**minimum daily temperature, the black line is 0°C and the colored lines are the dehardening and maximum level hardiness sensitivity experiments.**





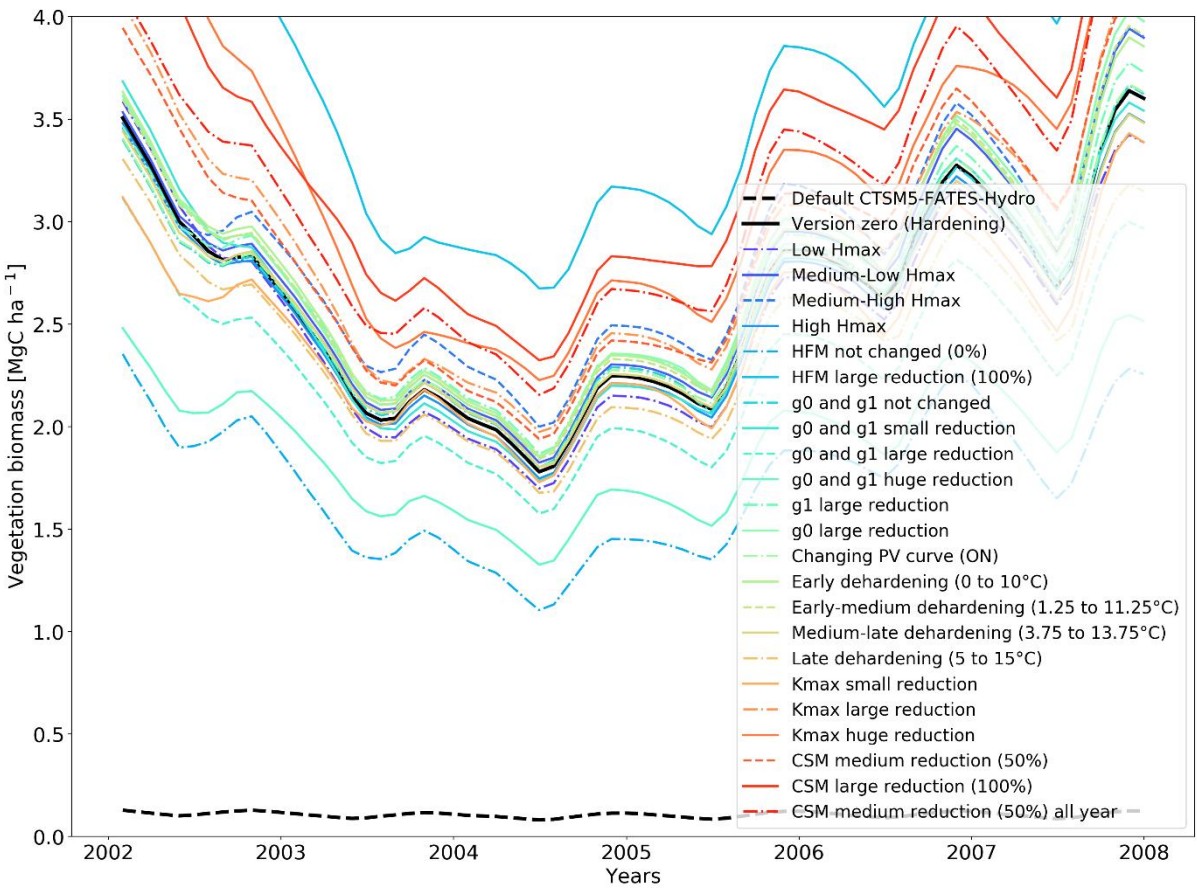

**Figure 11: Ensemble of living biomass simulations for needleleaf evergreen trees at the site of Spasskaya Pad during the period 2002-2008.**





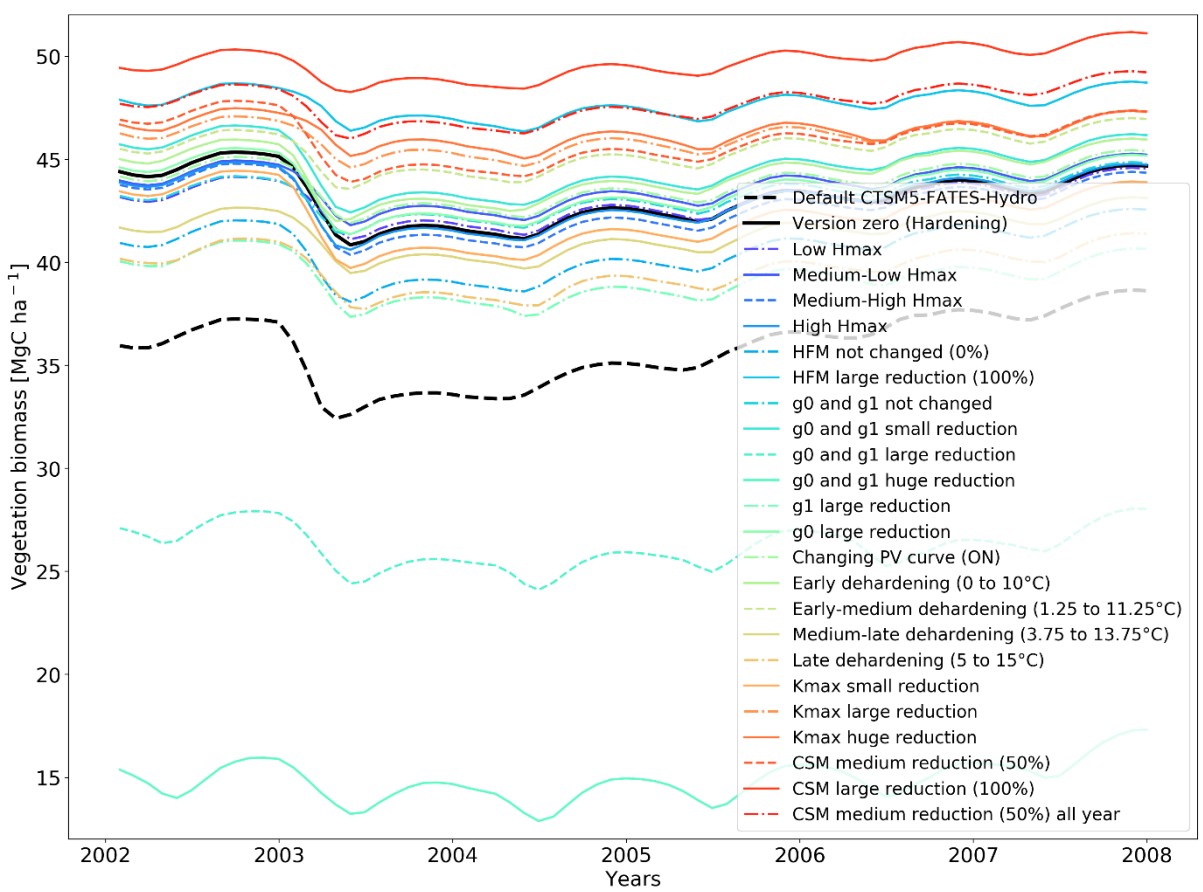

**Figure 12: Ensemble of living biomass simulations for needleleaf evergreen trees at the site of Farstanäs during the period 2002-2008.**

| Simulation | $K_{MAX}$ | $g0, g1$ | HFM | CSM | PV | *DR* | $H_{MAX}$ |
|---|---|---|---|---|---|---|---|
| Version zero (hardening) | $10^{\left(\frac{HD+3}{11}\right)}$ | $10^{\left(\frac{HD+3}{40}\right)}$ | 50% | 0% | OFF | 2.5-12.5°C | T5-10°C |

**Table 1: Configuration of the version zero simulation. This simulation was run at the sites of Spasskaya Pad and Farstanäs for evergreen needleleaf and deciduous broadleaf trees. T5 is the 5-year running mean of the minimum 2m daily temperature of each year.**

| $g0, g1$ | Not changed | Small $10^{\left(\frac{HD+3}{60}\right)}$ | Medium $10^{\left(\frac{HD+3}{40}\right)}$ | Large $10^{\left(\frac{HD+3}{20}\right)}$ | Huge $10^{\left(\frac{HD+3}{10}\right)}$ | Large only g0 $10^{\left(\frac{HD+3}{20}\right)}$ | Large only g1 $10^{\left(\frac{HD+3}{20}\right)}$ |
|---|---|---|---|---|---|---|---|
| *DR* | Early 0-10°C | Early-medium 1.25-11.25°C | Medium 2.5-12.5°C | Medium-late 3.75-13.75°C | Late 5-15°C | | |
| $H_{MAX}$ | High T5 - 5°C | High-medium T5 – 7.5°C | Medium T5 -10°C | Medium-low T5 -12.5°C | Low T5 - 15°C | | |





| $K_{MAX}$ | Small $10^{\left(\frac{HD+3}{13}\right)}$ | Medium $10^{\left(\frac{HD+3}{11}\right)}$ | Large $10^{\left(\frac{HD+3}{9}\right)}$ | Huge $10^{\left(\frac{HD+3}{7}\right)}$ |
|---|---|---|---|---|
| **CSM** | Small 0% | Medium 50% | Large 100% | Medium all year 50% |
| **HFM** | Small 0% | Medium 50% | Large 100% | |
| **PV** | OFF | ON | | |

**Table 2: Sensitivity experiments ran from the version zero hardening simulation (cells highlighted in orange). The simulations were run at the sites of Spasskaya Pad and Farstanäs for evergreen needleleaf trees. HD is the hardiness level and T5 is the 5-year running mean of the minimum 2m daily temperature of each year.**