# Peer review of "Inclusion of a cold hardening scheme to represent frost tolerance is essential to model realistic plant hydraulics in the Arctic-Boreal Zone in CLM5.0-FATES-Hydro"

_Geoscientific Model Development, 2022_

## Author Comment (AC1)

RC1: 'Comment on gmd-2022-136', Tim Artlip, 06 Sep 2022

This manuscript addresses the lack of a cold hardiness component in equations designed to assess water flow and fitness in landscape-level vegetation models, which don't work for low temperature climates, particularly with trees that potentially continue photosynthesis/ transpiration during winter.

The paper advances a logical addition of a cold hardiness component and includes novel data generated from simulations.

The logical addition of a cold hardiness component may not be considered a substantial advance by some readers.

The methods and assumptions are valid and clearly outlined. See also comments to the authors.

The results are sufficient the interpretations and conclusions. See also comments to the authors.

The model appears replicable by others in the field. See also comments to the authors.

The authors clearly state which contributions to the model are prior work by other authors and what their contributions are.

The title clearly indicates the contents of the manuscript including model name and number.

The abstract is clear and concise.

The presentation is well structured and clear.

The language is fluent and precise.

Mathematical formulae, symbols, abbreviations, and units are defined and precise.

The manuscript cannot be reduced.

The references are appropriate. See also comments to the authors.

The supplementary materials are sufficient and appropriate.

*Authors response: Thank you for the positive comments. Our responses are given in blue text under each comment.*

Comments to authors:

Line 175: I suggest the authors consider providing a supplementary file with these data (if possible).

*Authors response: Thank you for this comment. We added a table in the supplementary that includes the parameters used by Rammig et al. (2010) which were not specified in the paper (Please see Table S1 in the revised version of the manuscript).*

A crucial point is whether the authors will verify the model, particularly the hardiness estimations, with real experimental (biological) data as opposed to data from the literature.  While the simulations and conclusions look appropriate, I think confidence in the conclusions would be strengthened with a subsequent publication that tests the algorithms with actual biological data.

*Authors response: This is a good point, and it is mentioned in the conclusion and outlook of the manuscript. We highlight the need for plant hydraulic observations in cold regions, especially conductivities and fluxes in and out of the leaves/roots, but also the bulk elastic modulus and the*

*osmotic potential at full turgor, plant mortality due to hydraulic stress and carbon starvations, and various levels and thresholds useful to the hydro-hardening scheme. It would be very helpful to obtain more data on this and be able to evaluate and better constrain the hardening scheme and its effect on plant hydraulics. We do have a follow-up publication aimed at testing the role of cold hardening for vegetation survival and mortality during an extreme winter event in northern Norway, where we compare to observations of vegetation damage. Since that study frequently refers to this paper, we aim to submit that one once this manuscript has reached its final form.*

Line 175: The authors should consider the findings of Kuprian et al. (Tree Physiology 38, 591–601 doi:10.1093/treephys/tpx142) who examined the relationship between winter desiccation and bud primordia supercooling (hardiness) in Picea abies. Their results "suggest that there is no causal relationship between desiccation and the supercooling capacity of bud primordia in P. abies, but rather it involves other compounds within the cells of the bud primordium that reduce the water potential". This may be an important consideration in terms of biomass production as new needles arise from bud primordia.

*Authors response: Thank you for the reference. The findings by Kuprian et al. (2018) are indeed relevant and important to represent winter processes more accurately in models, and better estimate biomass changes. As explained by Kuprian et al., it seems, however, that there is no agreement yet on whether desiccation impacts supercooling or not (https://doi.org/10.1093/treephys/18.7.451). What most papers agree on is that desiccation decreases the risks of mechanical freezing damage by reducing the amount of water that may crystalize.*

*Here we added two extracts of Kuprian et al., (2018) where they explained their hypothesis on why/how the supercooling capacity of plants may be affected by frost-hardening:*

*"Total water potential of a tissue is determined by water content. Therefore, dehydration will decrease $\Psi_t$, but an osmotic component ($\Psi_o$) and cell wall turgor pressure ($\Psi_p$) also are contributing factors. Our results indicate that dehydration per se does not affect the supercooling capacity of bud primordia. Thus, the two other potentially relevant components of total water potential must be considered."*

*"A lesser cell wall turgor pressure ($\Psi_p$), resulting from alterations in cell wall elasticity, could also reduce $\Psi_t$ during winter. Seasonal changes in the composition of cell walls of bud primordia could also be possible, however cells in the primordia are composed of undifferentiated cells with only primary cell walls that are thin and not fully developed with potentially little leeway for changes in elasticity."*

*Based on these statements we cite their work at L264 in the revised version of the manuscript : "To maintain turgor during stress (Beck et al., 2007) or during hardening (Valentini et al., 1990), plant organs increase their solute concentration which decreases pinot and they increase the elasticity of their cell walls which corresponds to a decrease in epsil (Bartlett et al., 2012; Kuprian et al., 2018)."*

Figures: Many of the figures are difficult to make out as is the lettering within the figures. I suggest making the line/ symbols heavier and the colors bolder.

*Authors response: We thank you for this observation and suggestions. The figures have been made clearer by making the lines thicker and by putting the legends outside of the graphs in the revised version of the manuscript.*

---

## Author Comment (AC2)

In this manuscript, the authors attempted to incorporate the effect of cold hardening on the hydrological and physiological processes of trees into the CLM5.0-FATES-Hydro. The scheme of cold hardening consists of the hardening scheme (Rammig et al., 2010; some modifications) and the physiological scheme (maximum conductance, parameters for stomatal conductance, hydraulic failure mortality, pressure-volume curve, and carbon starvation mortality). They showed that the inclusions of cold hardening schemes are vital for reproducing the biomass of two boreal forests in Farstanas and Spasskaya Pad. Otherwise, the trees die due to hydraulic failure during the winter, caused by the low water potential of frozen soil and the resulting dehydration of the trees. Therefore, I think their schemes are successfully developed and valuable for many readers who want to model the processes in boreal forests.

*Authors response: We would like to thank the reviewer for this positive feedback on the manuscript. Please see our responses in blue texts under each comment.*

I recommend this article be accepted after the revisions listed below.

On the modifications to the scheme by Rammig et al. (2010)

For example, the authors modified the maximum hardiness level (H_MAX) from a constant (i.e., -30 deg C) by Rammig et al.(2010) to the variable changing with the running mean of the annual minimum air temperature of the past 5 years. This may result in a big difference in the simulations, particularly in Spasskaya Pad, but such a result is not shown in the present manuscript. I'd suggest showing the results when the original schemes by Rammig et al. (2010) are adopted so that the importance of the modifications in this study will be emphasized

*Authors response: Thank you for this suggestion. In principal it would make sense to test the scheme from Rammig et al. (2010) at Spasskaya Pad, but that scheme was only tested and parameterized for a location in Sweden, and not generalized to the rest of the world like our implementation. Due to the large climatic differences between Sweden and East Siberia, it is unlikely that the original scheme from Rammig et al. will perform well at Spasskaya Pad making it less valuable to compare to. This is especially true since the variables modified downstream of the hardening scheme ($K_{MAX}$, g0, g1 and HFM) have been adjusted based on $H_{MAX}$ and hence on the climate of a location. Still, the results from the test you suggested are partly represented by the sensitivity experiments on, for example, $H_{MAX}$. When we reduce T5 (the 5 year running mean of the minimum daily temperature) by 15°C instead of 10°C to define $H_{MAX}$, we end up with a reduced (less negative) hardiness level, which results in more dehydration but also more carbon uptake. In addition, we also compare the FATES model in its current state (default) with the FATES model containing the hydro-hardening scheme to show the importance of our model improvement. We have added more explanation to emphasize the necessity of the modifications in this study in the revised version of the manuscript: "Due to the large climatic differences between Sweden and East Siberia, it is unlikely that the original scheme from Rammig et al. will perform well at Spasskaya Pad. To deal with this, in our adaptation of the hardening model, $H_{MAX}$ becomes site- and time- dependent (to function globally and to account for evolution associated to changes in climate), and varies with the 5 year running mean of the annual minimum of daily mean air temperature at 2m height (T5)."*

Citations of equations

Throughout the manuscript, the citations of equations look strange and probably do not fit the style of GMD. In the case of this manuscript, all the "Eq. XX" should be put in parentheses. For example, in L182, "TH Eq. (1), HR Eq. (2) and DR Eq.(3)" should be "TH (Eq. (1)), HR (Eq. (2)), and DR (Eq.(3))". Please revise all of them.

*Authors response: Thank you for pointing this out. We have changed this for all equations based on some published papers in GMD where they do it like this: "(Eq. 1)". We also changed "Eq." to" Eqs." when appropriate.*

Symbols in equations

Throughout the manuscript, the symbols differ between the text and the equations. For example, H_MIN and H_MAX in the text are presented as M_min and H_max, respectively, in equations (1), (2), and (3). Please maintain the integrity of symbols.

*Authors response: Thank you for noticing this mismatch. $H_{MAX}$ and $H_{MIN}$ are now only mentioned this way.*

Section 2.2

I strongly suggest the authors provide the model description of HD (i.e., L191-202) in advance of those of TH, HR, and DR for better readability.

*Authors response: We agree that this improves readability, and we moved the text describing HD to L172 in the revised version of the manuscript, in advance of those of TH, HR, and DR:*

*Once a value has been assigned to TH, HR and DR, depending on the daily mean 2m air temperature, the model operates as follows: if TH is lower than the hardiness of the previous day (HDP), then HR is removed from HDP. By contrast, if TH is higher than HDP, DR is added to HDP (Eq. 1).*

$$HD = \begin{cases} HDP - HR, if\ HDP > TH \\ HDP + DR, if\ HDP \leq TH \end{cases}$$

Besides, the description L191-202 seems not sufficient. For example, HD takes the maximum value H_MIN in summer, but it is not explained. This corresponds to equations (1) and (3) of Rammig et al (2010). Here, Rammig et al. (2010) adopted aggd5 (the accumulated growing degree days), but the authors did not mention it in the manuscript. Is that OK?

*Authors response: We agree that the hardening scheme could be better explained and the differences with the Rammig scheme made clearer. Some changes were made in the manuscript and are described in the next response.*

Since this hardening scheme is the core of this study, the authors should describe it entirely, even if it is nearly the same as Rammig et al. (2010). Otherwise, the reader would have to refer to Rammig et al. (2010) when reading this paper.

*Authors response: Thank you for pointing this out. We have added 3 changes listed below in the revised version of the manuscript. We hope that this section is now less confusing and that all the modifications made to the scheme are better explained.*

*Added L1996: "In Rammig et al. (2010), the hardening period is prevented until the 210th Julian day and a growing degree day threshold is reached."*

*Added at L201: "In our version of the hardening scheme, if the requirements of Eq. (6) are met, the value given to HD in Eq. (1) will be overwritten."*

*Added at L205: "At the end of the time-step, values of HD outside of the range $H_{MIN}$ to $H_{MAX}$ will be redefined within these extremes according to Eq. (7).*

$$HD = \begin{cases} H_{MIN}, if\, HD > H_{MIN} \\ H_{MAX}, if\, HD < H_{MAX} \end{cases}$$
*(7)"*

In addition, according to equations (4) to (6), HD is determined depending on the interrelations between HDP and TH. I'd suggest showing the example of the temporal variation of HD and TH to show how these variables are interrelated.

*Authors response: According to the reviewer's comment, we have added Figure S14 in the Supplement to illustrate the evolution of TH and HD during two random years. Please also see the modification at L174 in the revised version of the manuscript: To illustrate Eq. (1) and the interrelation between HDP and HD, Figure S14 shows the temporal evolution of TH and HD during two random years.*

Figures

As reviewer 1 pointed out, most of the figures are difficult to distinguish between lines, and the legend obstructed the figure. Please try to make it easy to see, and put the legend outside the plot.

*Authors response: To make the figures clearer, the lines were made thicker on almost all graphs and large legends are placed outside of the graphs. The font size was also adjusted on most graphs.*

L61: It is unclear what "it" stands for here. Is it "plant hydraulics"? If so, I suggest rewriting this sentence as follows. "Plant hydraulics, apart from its critical role in the survival of plants during droughts, is also a major driver of species distribution."

*Authors response: Thank you for the suggestion. We have modified the sentence, please see L61 in the revised version of the manuscript. It now reads: "Plant hydraulics, apart from its critical role in the survival of plants during droughts, is also a major driver of species distribution."*

L110-111: Lawrence et al. (2019) --> (Lawrence et al., 2019)

*Authors response: Change made*

L178 and the caption of Table 1: Does "the minimum 2m daily temperature" mean the "annual minimum of daily mean air temperature at 2m height"? Describe it clearly.

*Authors response: "the minimum 2m daily temperature" was replaced by "the annual minimum of daily mean air temperature at 2m height"*

L192: "the hardiness of the previous day (HDP)" --> "the hardiness level of the previous day (HDP)" or "the HD of the previous day (HDP)"

*Authors response: "the hardiness of the previous day (HDP)" was replaced by "the HD of the previous day (HDP)"*

L245 (Eq. 12): The variables "HFMortScalar" and "percentage" appear here for the first time without any explanation. What are these?

*Authors response: Thank you for pointing this out. The variable "HFMortScalar" was replaced by "MortScalar" in (Eq. 12), and "percentage/100" was replaced by "50%". These changes should make the paragraph clearer and make the reader understand what we actually do to reduce the hydraulic failure mortality. We also added the two following sentences at L242 in the revised version of the manuscript: "In the control hardening simulation we reduced HFM by up to 50% at HD equal $H_{MAX}$ (see Eq. 12). In the two sensitivity experiments conducted on HFM, we modified the occurrences of 50% in Eq. (12) to obtain a reduction reaching 100% and 0% respectively (Table 1)."*

L262: Hd --> HD (italic)

*Authors response: Thank you for pointing this out. Change made.*

L342: 0.55% --> 55%

*Authors response: Thank you for the correction. Corrected to 55 %*

L342: (Fig. 8b and c) --> (Fig. 8a and b) Note that Fig. 8c shows the CSM, not HFM.

*Authors response: Thank you for pointing this out. "(Fig. 8b and c)" was replaced by (Fig. 8a and b) as suggested.*

L356: Insert "(Eq. (10))" to read "since HFM is a function of flc (Eq. (10))".

*Authors response: This was changed accordingly and "(Eq. 10)" was added at L360 in the revised version of the manuscript.*

L359-360: I could not get the meaning of this sentence. Does it mean "The contribution of the changes in K_MAX, g0, and g1 to the reduction of HFM can be seen by comparing Fig. 8b and Fig. S12."?

*Authors response: We have replaced "The contribution of the changes to $K_{MAX}$, g0 and g1, and the reduction of HFM, can be seen by comparing Fig. 8b and Fig. S12." with "The contribution of hardiness to the reduction of $K_{MAX}$, g0 and g1, and HFM, can be seen by comparing Fig. 8b and Fig. S12." in the revised version of the manuscript.*

L439: green --> red

*Authors response: Thank you. It's corrected.*

L439: brown --> green

*Authors response: Thank you. It's corrected.*

L440: dark green --> (light) blue

*Authors response: Corrected.*

Figures 8a and b, S5a and b, S12a and b, S13a and b: "Hydraulic mortality" in the vertical axis should be "Hydraulic failure mortality" to maintain the integrity of the terms.

*Authors response: Thank you. We have replaced "Hydraulic mortality" with "Hydraulic failure mortality" in Figures 8a and b, S5a and b, S12a and b, S13a and b.*